# Non-nucleoside reverse transcriptase inhibitor-based combination antiretroviral therapy is associated with lower cell-associated HIV RNA and DNA levels compared to protease inhibitor-based therapy

Alexander O Pasternak[1]*, Jelmer Vroom[1], Neeltje A Kootstra[2], Ferdinand WNM Wit[3,4,5,6], Marijn de Bruin[7,8], Davide De Francesco[9], Margreet Bakker[1], Caroline A Sabin[9], Alan Winston[10], Jan M Prins[6], Peter Reiss[3,4,5], Ben Berkhout[1], The Co-morBidity in Relation to Aids (COBRA) Collaboration

[1]Amsterdam UMC, University of Amsterdam, Laboratory of Experimental Virology, Department of Medical Microbiology and Infection Prevention, Amsterdam, Netherlands; [2]Amsterdam UMC, University of Amsterdam, Laboratory of Viral Immune Pathogenesis, Department of Experimental Immunology, Amsterdam, Netherlands; [3]Amsterdam Institute for Global Health and Development, Amsterdam, Netherlands; [4]Amsterdam UMC, University of Amsterdam, Department of Global Health, Amsterdam Institute for Infection and Immunity, Amsterdam, Netherlands; [5]HIV Monitoring Foundation, Amsterdam, Netherlands; [6]Amsterdam UMC, University of Amsterdam, Department of Internal Medicine, Amsterdam, Netherlands; [7]Health Psychology Group, Institute of Applied Health Sciences, University of Aberdeen, Aberdeen, United Kingdom; [8]Radboud University Medical Center, Radboud Institute for Health Sciences, Nijmegen, Netherlands; [9]Institute for Global Health, University College London, London, United Kingdom; [10]Department of Medicine, Imperial College London, London, United Kingdom

*For correspondence:
a.o.pasternak@amsterdamumc.nl

Group author details:
The Co-morBidity in Relation to Aids (COBRA) Collaboration See page 14

Competing interests: The authors declare that no competing interests exist.

## Abstract

**Background:** It remains unclear whether combination antiretroviral therapy (ART) regimens differ in their ability to fully suppress human immunodeficiency virus (HIV) replication. Here, we report the results of two cross-sectional studies that compared levels of cell-associated (CA) HIV markers between individuals receiving suppressive ART containing either a non-nucleoside reverse transcriptase inhibitor (NNRTI) or a protease inhibitor (PI).

**Methods:** CA HIV unspliced RNA and total HIV DNA were quantified in two cohorts (n = 100, n = 124) of individuals treated with triple ART regimens consisting of two nucleoside reverse transcriptase inhibitors (NRTIs) plus either an NNRTI or a PI. To compare CA HIV RNA and DNA levels between the regimens, we built multivariable models adjusting for age, gender, current and nadir CD4$^+$ count, plasma viral load zenith, duration of virological suppression, NRTI backbone composition, low-level plasma HIV RNA detectability, and electronically measured adherence to ART.

**Results:** In both cohorts, levels of CA HIV RNA and DNA strongly correlated (rho = 0.70 and rho = 0.54) and both markers were lower in NNRTI-treated than in PI-treated individuals. In the

multivariable analysis, CA RNA in both cohorts remained significantly reduced in NNRTI-treated individuals ($p_{adj}$ = 0.02 in both cohorts), with a similar but weaker association between the ART regimen and total HIV DNA ($p_{adj}$ = 0.048 and $p_{adj}$ = 0.10). No differences in CA HIV RNA or DNA levels were observed between individual NNRTIs or individual PIs, but CA HIV RNA was lower in individuals treated with either nevirapine or efavirenz, compared to PI-treated individuals.

**Conclusions:** All current classes of antiretroviral drugs only prevent infection of new cells but do not inhibit HIV RNA transcription in long-lived reservoir cells. Therefore, these differences in CA HIV RNA and DNA levels by treatment regimen suggest that NNRTIs are more potent in suppressing HIV residual replication than PIs, which may result in a smaller viral reservoir size.

**Funding:** This work was supported by ZonMw (09120011910035) and FP7 Health (305522).

## Introduction

In individuals who are able to adhere to combination antiretroviral therapy (ART), therapy suppresses human immunodeficiency virus (HIV) replication, restores immune function, and prevents the development of acquired immunodeficiency syndrome (AIDS) (*Deeks et al., 2013*). More than 20 different antiretroviral drugs belonging to six main classes are currently approved for clinical use (*Pau and George, 2014*). Depending on the class, these drugs block different steps of the HIV replication cycle, such as reverse transcription, proviral integration, or virus particle maturation. Current ART regimens typically consist of two nucleotide or nucleoside reverse transcriptase inhibitors (NRTIs) and a third drug of another class, for example, a non-nucleoside reverse transcriptase inhibitor (NNRTI), a protease inhibitor (PI), or an integrase strand transfer inhibitor (INSTI).

Despite the efficient suppression of HIV replication, ART is not curative and has to be sustained lifelong. Persistence of viral reservoirs forms the major obstacle to an HIV cure (*Deeks et al., 2016*). Viral reservoir markers, such as low-level HIV RNA in plasma (residual viremia) and cell-associated (CA) HIV DNA and RNA, can be measured in most treated individuals with plasma HIV RNA suppressed to below the limit of quantification of commercial assays (*Jacobs et al., 2019*; *Pasternak and Berkhout, 2018*; *Pasternak et al., 2013*; *Avettand-Fènoël et al., 2016*; *Sharaf and Li, 2017*). Although HIV latent reservoirs persist primarily by cell longevity and proliferation (*Chomont et al., 2009*; *Maldarelli et al., 2014*; *Wagner et al., 2014*), replenishment of the reservoirs by residual virus replication despite ART has been proposed as an alternative mechanism of HIV persistence (*Chun et al., 2005*; *Lorenzo-Redondo et al., 2016*; *Sigal et al., 2011*). The latter possibility remains a matter of longstanding debate in HIV research field (*Martinez-Picado and Deeks, 2016*). Residual HIV replication can result from insufficient penetration of antiretroviral drugs into tissues and anatomic sanctuaries, causing reduced local drug concentrations in tissue sites (*Estes et al., 2017*; *Fletcher et al., 2014*). However, most (but not all) studies could not demonstrate any measurable virus evolution in peripheral blood and tissues of ART-treated individuals (*Bozzi et al., 2019*; *Van Zyl et al., 2017*; *Joos et al., 2008*). This lack of significant virus evolution on ART is considered one of the strongest arguments against residual HIV replication. On the other hand, a transient increase in episomal HIV DNA has been demonstrated in a number of trials of ART intensification with raltegravir, an INSTI (*Buzón et al., 2010*; *Hatano et al., 2013a*). This accumulation of unintegrated HIV DNA, observed upon blocking integration, revealed ongoing integration events prior to intensification. Because all other antiretroviral drug classes act upstream of INSTIs, this implies that complete rounds of HIV replication (infection of new cells) had been ongoing pre-intensification. However, no decrease in residual HIV viremia could be demonstrated in these and other intensification trials (*Gandhi et al., 2010*; *McMahon et al., 2010*; *Hatano et al., 2011*).

It is also a matter of debate whether different ART regimens are equally potent in suppressing residual HIV replication. All current antiretroviral drugs act by preventing the infection of new cells and are not expected to inhibit HIV RNA transcription or virus production in the long-lived reservoir cells that were infected prior to ART initiation, or in the progeny of such cells. Therefore, if regimens are equally potent in stopping the infection of new cells, one would not expect to detect a difference in residual viremia or CA RNA levels between ART regimens. Consequently, finding such a difference would suggest that some regimens are more potent in suppressing residual replication than others, arguing that virus suppression is less complete with at least some of the regimens. A number of studies reported lower levels of residual viremia in individuals treated with NNRTI-based,

compared to PI-based, ART regimens (*Geretti et al., 2019*; *Darcis et al., 2020a*, reviewed in *Darcis et al., 2020b*). However, to date, few studies have compared levels of CA HIV reservoir markers between different ART regimens (*Nicastri et al., 2008*; *Sarmati et al., 2012*; *Kiselinova et al., 2015*). Here, we cross-sectionally measured CA HIV RNA and DNA in two cohorts of individuals receiving suppressive ART containing two NRTIs and either an NNRTI or a PI.

# Materials and methods

## Key resources table

| Reagent type (species) or resource | Designation | Source or reference | Identifiers | Additional information |
|---|---|---|---|---|
| Biological sample (*Homo sapiens*) | PBMC samples from HIV-infected individuals | | | |
| Commercial assay or kit | DNA-free DNA Removal Kit | ThermoFisher Scientific | Cat# AM1906 | |
| Commercial assay or kit | Platinum Quantitative PCR SuperMix-UDG | ThermoFisher Scientific | Cat# 11730–025 | |
| Commercial assay or kit | TaqMan β-Actin Detection Reagents | ThermoFisher Scientific | Cat# 401846 | |
| Commercial assay or kit | TaqMan Ribosomal RNA Control Reagents | ThermoFisher Scientific | Cat# 4308329 | |
| Chemical compound, drug | SuperScript III reverse transcriptase | ThermoFisher Scientific | Cat# 18080–085 | |
| Chemical compound, drug | Random primers | ThermoFisher Scientific | Cat# 48190–011 | |
| Chemical compound, drug | RNaseOUT Recombinant Ribonuclease Inhibitor | ThermoFisher Scientific | Cat# 10777–019 | |
| Software, algorithm | Prism 8.3.0 | GraphPad Software | https://www.graphpad.com/; RRID:SCR_002798 | |
| Software, algorithm | IBM SPSS Statistics (version 25) | IBM Corporation | https://www.ibm.com/; RRID:SCR_019096 | |

## Study participants

Participants for the COmorBidity in Relation to AIDS (COBRA) cohort were recruited at two clinical sites in Amsterdam (The Netherlands) and London (UK) from ongoing prospective cohort studies on co-morbidity and aging in HIV, the AGEhIV Cohort Study in Amsterdam (*Schouten et al., 2014*), and the POPPY study in London (*De Francesco et al., 2016*). All participants were required to be at least 45 years of age. The study design and participant characteristics were reported previously (*De Francesco et al., 2018*). Although most COBRA participants had two study visits within 2 years, for the present cross-sectional study, only peripheral blood mononuclear cell (PBMC) samples from the first study visit were used: 63 participants out of 100 were from the Amsterdam sub-cohort and 37 were from the London sub-cohort. The COBRA study was approved by the institutional review board of the Academic Medical Center (Medisch Ethische Toetsingscommissie, reference number NL 30802.018.09) and a UK Research Ethics Committee (REC) (reference number 13/LO/0584 Stanmore, London). All participants provided written informed consent.

Participants for the Adherence Improving Self-Management Strategy (AIMS) randomized trial were recruited at the HIV outpatient clinic of the Academic Medical Center (Amsterdam). Adherence to ART in this cohort was measured electronically using Medication Event Monitoring System (MEMS)-cap pill bottles (Aardex, Switzerland), which record the moments of bottle opening. Adherence was defined as the percentage of doses taken within a specified time interval (11–13 hr for

twice-daily and 22–26 hr for once-daily regimens) during the assessment period. For the present study, the adherence was assessed during 1-month periods that finished less than 20 days before or after the HIV sampling moments. The randomized trial, the results of which were reported previously (*de Bruin et al., 2010*), assessed the impact of a behavioral intervention to increase adherence; therefore, adherence data and PBMC samples were collected at several time points. For the present cross-sectional study, for 93.3% of the participants we used the 'baseline' (pre-randomization) PBMC samples and the corresponding adherence assessment data. For the remaining 6.6%, who lacked baseline PBMC samples, samples and data from the subsequent time point were used. The AIMS study was approved by the institutional review board of the Academic Medical Center (protocol number NTR176). The trial is registered at https://www.isrctn.com (ISRCTN97730834). All participants provided written informed consent.

Historical plasma HIV RNA measurements, CD4+ T-cell counts, and treatment data were retrieved from the outpatient medical records. The duration of continuous virological suppression was calculated as the duration of the latest period with undetectable plasma HIV RNA prior to the measurement, allowing isolated 'blips' of 50–999 copies/ml. The duration of cumulative suppression was calculated by adding together all such periods of continuous suppression. The duration of the current regimen was calculated as the period during which the participant had been receiving combination ART that included their current NNRTI or PI drug and no other NNRTI or PI.

## Virological measurements

Plasma HIV RNA was measured using commercial assays with detection limits of 40 or 50 copies/ml. For CA HIV RNA and total HIV DNA measurements, total nucleic acids were extracted from PBMC using the Boom isolation method (*Boom et al., 1990*). Extracted cellular RNA was treated with DNase (DNA-free kit; Thermo Fisher Scientific) to remove DNA that could interfere with the quantitation and reverse transcribed using random primers and SuperScript III reverse transcriptase (all from Thermo Fisher Scientific). CA HIV unspliced RNA and total HIV DNA were measured using previously described quantitative polymerase chain reaction (qPCR)-based methods (*Pasternak et al., 2008*; *Malnati et al., 2008*). HIV DNA or RNA copy numbers were determined using a 7-point standard curve with a linear range of >5 orders of magnitude that was included in every qPCR run and normalized to the total cellular DNA (by measurement of β-actin DNA) or RNA (by measurement of 18S ribosomal RNA) inputs, respectively, as described previously (*Pasternak et al., 2009*). Non-template control wells were included in every qPCR run and were consistently negative. Total HIV DNA was detectable in 90.0% of participants in the COBRA cohort and in 87.8% in the AIMS cohort. CA HIV RNA was detectable in 86.9% of participants in the COBRA cohort and in 83.7% in the AIMS cohort. Undetectable measurements of CA RNA or DNA were assigned the values corresponding to 50% of the corresponding assay detection limits. The detection limits depended on the amounts of the normalizer (input cellular DNA or RNA), and therefore differed between samples.

## Statistical analysis

Variables were compared between NNRTI- and PI-based ART by using Mann-Whitney tests for continuous variables and Fisher's exact tests or Chi-square tests for categorical variables. Strength of the associations between CA RNA or DNA and other variables was initially assessed by nonparametric Spearman or Mann-Whitney tests, as appropriate, and subsequently by fitting generalized linear models (GLMs) on rank-transformed dependent variables. Binary explanatory variables were included in the models if the representation of the least frequent category was >5%. Therefore, gender and plasma HIV RNA detectability were not included in the model in the COBRA cohort. Similarly, a threshold of 5% was used for inclusion of the NRTI backbone categories in the analysis, resulting in the inclusion of three most frequent categories for each cohort. The most frequent NRTI backbone category was used as a reference category. Explanatory variables that were associated with the dependent variables with a sufficient strength ($p < 0.1$) in univariable GLM analyses were included in multivariable models. Individual tests are described in the legends to figures and tables. Data were analyzed using Prism 8.3.0 (GraphPad Software) and IBM SPSS Statistics (version 25). All tests were two-sided. p-values <0.05 were considered statistically significant.

## Results

### CA HIV RNA and DNA in the COBRA cohort

We measured CA HIV unspliced RNA and total HIV DNA in PBMC samples from participants of the COBRA cohort (*De Francesco et al., 2018*). COBRA is a cohort of HIV-infected individuals aged 45 years or older with sustained HIV suppression on ART recruited from two large European HIV treatment centers in Amsterdam and London. Of the 132 COBRA participants with available PBMC samples, 100 were treated with ART that consisted of two NRTIs plus either one NNRTI (n = 58) or one ritonavir-boosted PI (n = 42) and were included in the analysis. Samples were obtained between

**Table 1.** Characteristics of participants treated with NNRTI- and PI-based ART regimens.

| Variable | | COBRA cohort (n = 100) | | | AIMS cohort (n = 124) | | |
|---|---|---|---|---|---|---|---|
| | | NNRTI (n = 58) | PI (n = 42) | p[i] | NNRTI (n = 88) | PI (n = 36) | p |
| Age (years) | | 55 (51–61)[ii] | 56 (50–62) | 0.97 | 47 (41–54) | 44 (39–53) | 0.23 |
| Male gender | | 56 (96.6) | 39 (92.9) | 0.65 | 78 (88.6) | 31 (86.1) | 0.76 |
| Current CD4$^+$ count (cells/mm$^3$) | | 640 (511–796) | 617 (408–782) | 0.21 | 550 (368–798) | 575 (470–745) | 0.46 |
| CD4$^+$ count nadir (cells/mm$^3$) | | 180 (115–253) | 200 (88–253) | 0.91 | 160 (83–240) | 165 (85–220) | 0.78 |
| Plasma HIV RNA zenith (log$_{10}$ copies/ml) | | 5.08 (4.71–5.52) | 5.00 (4.72–5.70) | 0.84 | 5.21 (4.68–5.62) | 5.35 (4.96–5.97) | 0.09 |
| Duration of cumulative virological suppression (months) | | 137.0 (93.3–171.3) | 90.4 (46.5–133.1) | 0.004 | 55.6 (28.5–90.2) | 40.2 (11.0–87.5) | 0.20 |
| Duration of continuous virological suppression (months) | | 118.3 (73.6–151.6) | 62.2 (33.5–118.4) | 0.001 | 45.4 (25.4–74.3) | 19.9 (6.2–64.3) | 0.01 |
| Duration of the current NNRTI or PI regimen (months) | | 98.6 (48.4–136.6) | 48.0 (26.3–68.2) | <0.0001 | 39.1 (12.4–59.9) | 12.5 (7.2–23.6) | <0.0001 |
| Current plasma HIV RNA <50 copies/ml[iii] | | 56 (96.6) | 42 (100.0) | 0.51 | 80 (90.9) | 27 (75.0) | 0.04 |
| Adherence to ART (%)[iv] | | - | - | - | 89.2 (63.6–100) | 91.3 (65.5–100) | 0.55 |
| NRTI backbone | FTC + TDF[v] | 47 (81.0) | 31 (73.8) | 0.43 | 8 (9.1) | 4 (11.1) | 0.26 |
| | ABC + 3TC | 4 (6.9) | 6 (14.3) | | 4 (4.5) | - | |
| | 3TC + TDF | 5 (8.6) | 2 (4.8) | | 42 (47.7) | 21 (58.3) | |
| | 3TC + AZT | 2 (3.4) | 1 (2.4) | | 26 (29.5) | 6 (16.7) | |
| | Other[vi] | - | 2 (4.8) | | 8 (9.1) | 5 (13.9) | |
| NNRTI | EFV[vii] | 28 (48.3) | - | | 46 (52.3) | - | |
| | NVP | 26 (44.8) | - | | 41 (46.6) | - | |
| | Other[viii] | 4 (6.9) | - | | 1 (1.1) | - | |
| PI | ATZ/r[ix] | - | 19 (45.2) | | - | 22 (61.1) | |
| | DRV/r | - | 16 (38.1) | | - | - | |
| | LPV/r | - | 3 (7.1) | | - | 9 (25.0) | |
| | Other[x] | - | 4 (9.5) | | - | 5 (13.9) | |

[i]Mann-Whitney tests were used for continuous variables and Fisher's exact tests or Chi-square tests were used for categorical variables.

[ii]Data are medians (interquartile ranges) for continuous variables and numbers (percentages) for discrete variables.

[iii]Where detectable, plasma HIV RNA was <400 copies/ml for all patients.

[iv]Adherence was measured electronically.

[v]NRTIs: FTC, emtricitabine; TDF, tenofovir disoproxil fumarate; ABC, abacavir; 3TC, lamivudine; AZT, zidovudine; D4T, stavudine; DDI, didanosine.

[vi]COBRA: ABC+TDF – 1 (PI), ABC+AZT – 1 (PI). AIMS: 3TC+D4T – 3 (NNRTI), 3TC+DDI – 3 (NNRTI) + 2 (PI), D4T+DDI – 1 (NNRTI), DDI+TDF – 1 (NNRTI) + 1 (PI), 3TC+FTC – 1 (PI), AZT+DDI – 1 (PI).

[vii]NNRTIs: EFV, efavirenz; ETR, etravirine; NVP, nevirapine; RIL, rilpivirine.

[viii]COBRA: ETR – 2, RIL – 2. AIMS: unknown – 1.

[ix]PIs: ATZ, atazanavir; DRV, darunavir; FOS, fosamprenavir; LPV, lopinavir; SAQ, saquinavir; IDV, indinavir; /r, ritonavir-boosted PI.

[x]COBRA: FOS/r – 3, SAQ/r – 1. AIMS: ATZ – 3, IDV/r – 1, IDV – 1.

April 2011 and December 2014. *Table 1* shows the participant characteristics, grouped according to the treatment regimen. In brief, 95% were male and the median age was 55 years (interquartile range, 51–61 years). 98 participants had undetectable plasma HIV RNA (<50 copies/ml) and two had detectable but low levels (66 and 90 copies/ml). Participants had a median of 118 (62–163) months of cumulative and 99 (47–146) months of continuous virological suppression on ART prior to the measurements and had been treated with their current NNRTI or PI regimen for a median of 69 (38–116) months. The duration of virological suppression on ART and the duration of current regimen prior to the measurements were significantly different between NNRTI- and PI-treated participants (cumulative suppression: median of 137 vs 90 months, respectively, p = 0.004; continuous suppression, median of 118 vs 62 months, respectively, p = 0.001; current regimen: median of 99 vs 48 months, respectively, p < 0.0001).

The median CA HIV RNA and total HIV DNA levels in the COBRA cohort were 2.15 (1.58–2.52) $\log_{10}$ copies/µg total RNA and 2.50 (1.84–2.77) $\log_{10}$ copies/$10^6$ PBMC, respectively. *Figure 1A* shows correlations of current CD4+ count, CD4+ count nadir, plasma HIV RNA zenith, and duration of continuous virological suppression prior to the measurements, with CA HIV RNA and DNA. Significant correlations with both HIV RNA and DNA were observed for the plasma HIV RNA zenith (rho = 0.22, p = 0.04 and rho = 0.36, p = 0.0004, respectively), but not for any other variable. Duration of cumulative virological suppression and duration of the current regimen were also not associated with either CA HIV RNA or DNA (cumulative suppression: rho = 0.04, p = 0.68 and rho = -0.06, p = 0.60; current regimen: rho = 0.02, p = 0.82 and rho = 0.02, p = 0.87). Furthermore, CA RNA and DNA strongly correlated (rho = 0.70, p < 0.0001) and both markers were lower in NNRTI- than in PI-treated participants (CA RNA: 1.78 (1.58–2.29) vs 2.36 (1.55–2.65) $\log_{10}$ copies/µg total RNA, p = 0.03; total DNA: 2.46 (1.78–2.64) vs 2.60 (1.93–2.90) $\log_{10}$ copies/$10^6$ PBMC, p = 0.07).

To assess the association of CA HIV RNA and DNA with ART regimens, we built multivariable GLMs, adjusted for a number of demographic and clinical variables (*Figure 2A*, *Supplementary file 1a*). Higher plasma HIV RNA zenith and PI-based ART regimen remained significantly associated with higher levels of both CA HIV RNA and DNA in the multivariable analysis (plasma HIV RNA zenith: $p_{adj}$ = 0.02 and $p_{adj}$ = 0.0001, respectively; ART regimen: $p_{adj}$ = 0.02 and $p_{adj}$ = 0.048, respectively).

## CA HIV RNA and DNA in the AIMS cohort

Having established an association between CA HIV RNA and DNA and the ART regimen in the COBRA cohort, we sought to validate these observations in another cohort. To this end, we used PBMC samples from participants of the AIMS randomized controlled trial that investigated the effects of a behavioral intervention to increase adherence to ART (*de Bruin et al., 2010*). Participants for this trial with electronically measured adherence had been recruited from HIV-infected individuals on ART visiting the outpatient clinic of the Academic Medical Center (Amsterdam, The Netherlands). Samples were obtained between March 2005 and February 2007. Of the 147 AIMS participants with available PBMC samples, 124 were treated with ART that consisted of two NRTIs plus either one NNRTI (n = 88) or one PI (n = 36) and were included in the analysis. *Table 1* shows the participant characteristics. In brief, 88% were male and the median age was 46 years (interquartile range, 40–54 years). Median adherence to ART was 91% (66–100%). 107 out of 124 participants had undetectable plasma HIV RNA (<50 copies/ml) and 17 had detectable but low levels (range, 52–366 copies/ml). The duration of cumulative (median, 47 (25–88) months) and continuous (40 (17–72) months) virological suppression on ART prior to the measurements, as well as the duration of current NNRTI or PI regimen (median, 26 (10–46) months), was shorter in the AIMS compared to the COBRA cohort. As in the COBRA cohort, the duration of continuous virological suppression on ART prior to the measurements and the duration of the current regimen were significantly different between NNRTI- and PI-treated participants (continuous suppression, medians of 45 vs 20 months, respectively, p = 0.01; current regimen, medians of 39 vs 13 months, respectively, p < 0.0001). In addition, low-level plasma HIV RNA was detectable more frequently in PI-treated than in NNRTI-treated participants (25.0% vs 9.1%, p = 0.04). Other variables, including adherence to ART, did not differ between NNRTI- and PI-treated participants.

The median CA HIV RNA and total HIV DNA levels in the AIMS cohort were 1.71 (1.25–2.01) $\log_{10}$ copies/µg total RNA and 2.41 (1.88–2.79) $\log_{10}$ copies/$10^6$ PBMC, respectively. *Figure 1B* shows correlations of current CD4+ count, CD4+ count nadir, plasma HIV RNA zenith, and duration

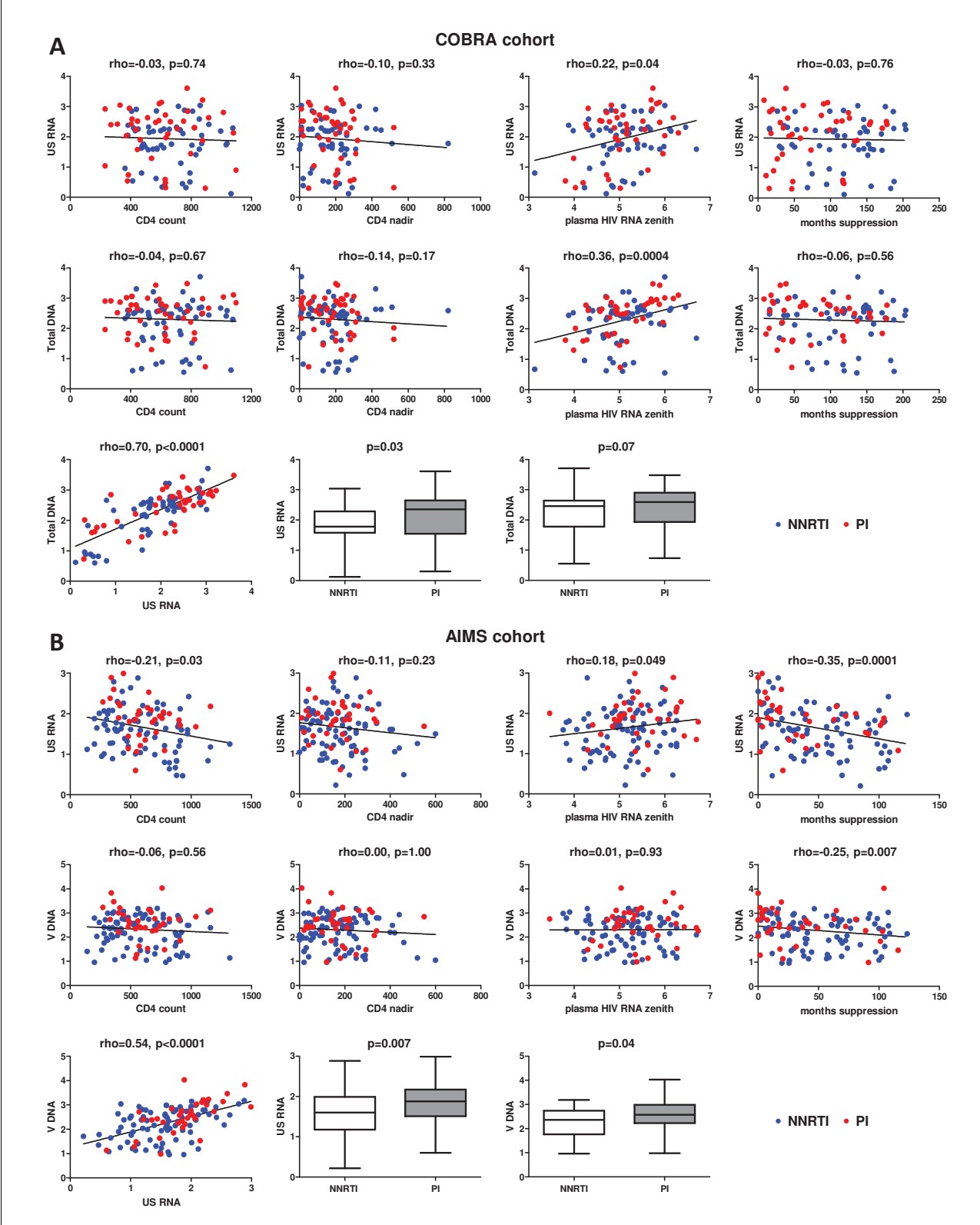

**Figure 1.** Associations of clinical and virological variables, time of virological suppression, and ART regimens (NNRTI-based vs PI-based) with the levels of cell-associated human immunodeficiency virus (HIV) unspliced RNA (US RNA) and total HIV DNA in (**A**) COBRA cohort (n = 100) and (**B**) AIMS cohort (n = 124). Units of measurement are US RNA: $\log_{10}$ copies/μg total RNA, total DNA: $\log_{10}$ copies/$10^6$ peripheral blood mononuclear cells (PBMC), CD4 count and CD4 nadir: cells/mm$^3$, plasma HIV RNA zenith: $\log_{10}$ copies/ml. Levels of significance were calculated by Spearman correlation analyses or

*Figure 1 continued on next page*

Figure 1 continued

Mann-Whitney tests, as appropriate. In all correlation graphs, non-nucleoside reverse transcriptase inhibitor (NNRTI)- and protease inhibitor (PI)-treated participants are color-coded (NNRTI - blue, PI - red).

The online version of this article includes the following figure supplement(s) for figure 1:

**Figure supplement 1.** Effects of duration of cumulative virological suppression and duration of the current regimen on the levels of cell-associated HIV US RNA and total HIV DNA in the AIMS cohort (n = 124).

**Figure supplement 2.** Differences in duration of continuous and cumulative virological suppression, duration of current regimen, and in adherence to ART between participants with undetectable vs low-level detectable pVL in the AIMS cohort.

of continuous virological suppression prior to the measurements with CA HIV RNA and DNA. In contrast to the COBRA cohort, duration of continuous virological suppression was significantly negatively associated with both CA HIV RNA (rho = -0.35, p = 0.0001) and total HIV DNA (rho = -0.25, p = 0.007). Duration of cumulative virological suppression and duration of the current regimen were also significantly negatively associated with both CA HIV RNA (rho = -0.26, p = 0.004 and rho = -0.23, p = 0.01, respectively) and total HIV DNA (rho = -0.20, p = 0.03 and rho = -0.18, p = 0.04, respectively), but these associations were weaker than those of the duration of continuous suppression (*Figure 1—figure supplement 1*). In addition, current CD4+ count and plasma HIV RNA zenith were negatively (rho = -0.21, p = 0.03) and positively (rho = 0.18, p = 0.049) associated with CA RNA but not with total DNA (*Figure 1B*). As in the COBRA cohort, CA RNA and DNA strongly correlated (rho = 0.54, p < 0.0001) and were lower in NNRTI- than in PI-treated participants (CA RNA: 1.60 (1.18–1.99) vs 1.88 (1.51–2.17) $\log_{10}$ copies/μg total RNA, p = 0.007; total DNA: 2.36 (1.76–2.74) vs 2.57 (2.22–2.98) $\log_{10}$ copies/$10^6$ PBMC, p = 0.04).

Next, we built multivariable GLMs to assess the association of CA HIV RNA and DNA with ART regimens in the AIMS cohort (*Figure 2B*, *Supplementary file 1b*). In addition to the same variables as for the COBRA cohort, these models included gender, plasma HIV RNA detectability, and adherence to ART. Due to co-linearity between the durations of continuous and cumulative virological suppression and the duration of current regimen, only duration of continuous suppression was included in the multivariable analysis, as its associations with HIV RNA and DNA were the strongest among these three measures. The shorter duration of continuous virological suppression prior to the measurements and PI-based ART regimen remained significantly associated with higher levels of CA HIV RNA in the multivariable analysis (duration of suppression: $p_{adj}$ = 0.04; ART regimen: $p_{adj}$ = 0.02). The shorter duration of continuous suppression was also significantly associated with higher total HIV DNA ($p_{adj}$ = 0.03), while the association of ART regimen with HIV DNA did not achieve statistical significance ($p_{adj}$ = 0.10). We also built three alternative models, in which either duration of cumulative suppression or the duration of current regimen was included instead of the duration of continuous suppression, or the duration of continuous suppression was included together with the duration of current regimen. The adjusted associations of CA HIV RNA with the ART regimen remained significant in these alternative models (*Figure 2—figure supplement 1*, *Figure 2—figure supplement 2*).

## Sensitivity analysis and associations of individual antiretroviral drugs with CA HIV RNA and DNA in the pooled cohort

Having observed similar associations of the ART regimen with CA HIV RNA and DNA in both COBRA and AIMS cohorts, we pooled the two cohorts in order to achieve sufficient statistical power to perform a sensitivity analysis and to assess the associations of individual antiretroviral drugs with the levels of CA HIV RNA and DNA. As 11 individuals participated in both cohorts 7 years apart, we excluded the second samples of these individuals from the analysis, bringing the total number of participants in the pooled cohort to 213.

As expected, both CA HIV RNA and DNA were significantly lower in NNRTI- than in PI-treated participants of the pooled cohort (p = 0.0006 and p = 0.01, respectively) (*Figure 3A*). In accordance with this, CA RNA/DNA ratios were not significantly different by ART regimen, although a trend was observed toward lower CA RNA/DNA ratios in NNRTI-treated participants (p = 0.19) (*Figure 3—figure supplement 1*). To demonstrate that the associations of ART regimens with CA RNA and DNA also hold in those individuals who are stably suppressed on therapy, we performed a sensitivity analysis, limiting the analysis to a subset of participants with undetectable plasma HIV RNA and more than 6 months of continuous virological suppression on ART (n = 178). In this subset, CA HIV RNA

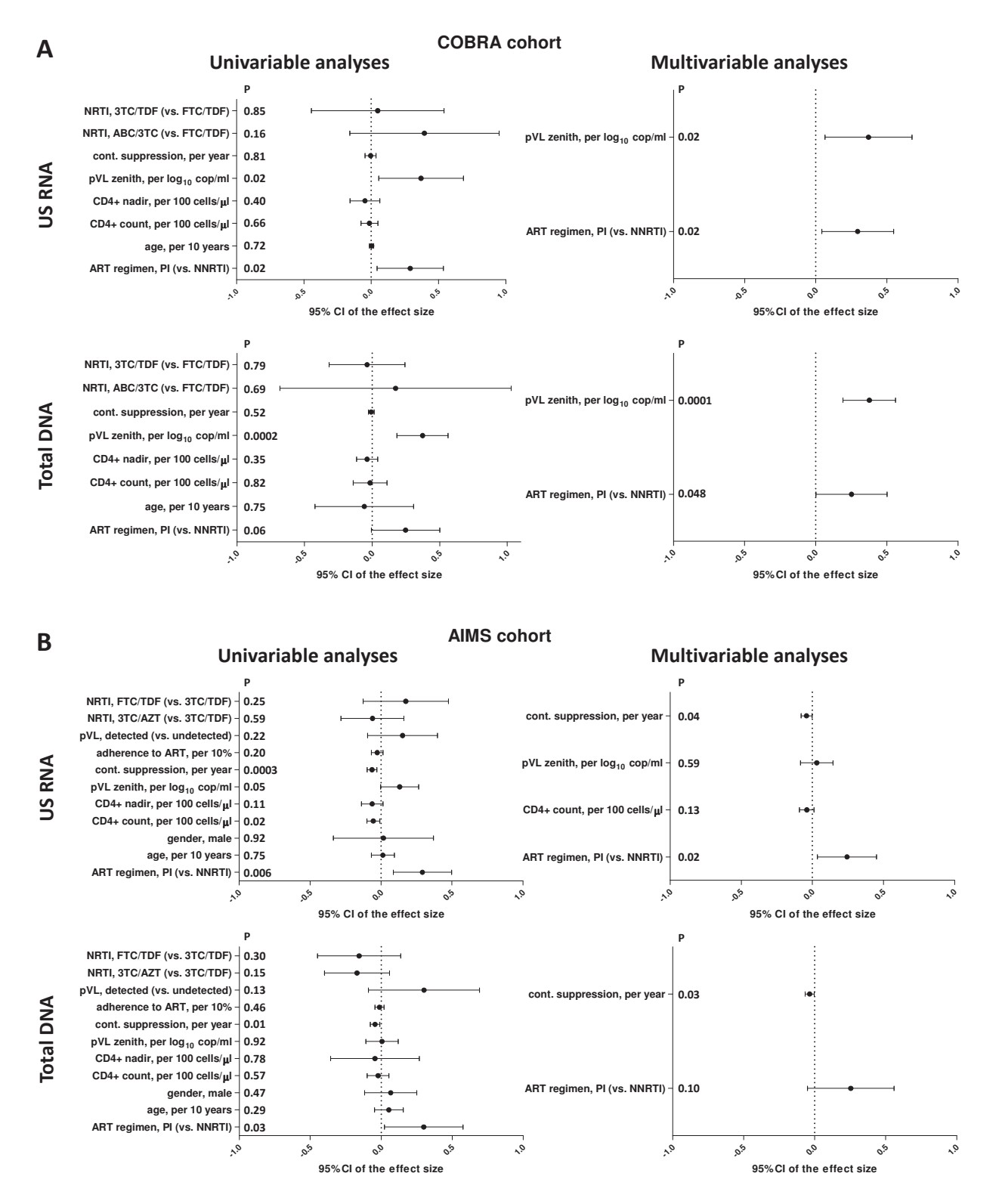

**Figure 2.** Regression analyses to identify variables associated with cell-associated human immunodeficiency virus (HIV) unspliced (US) RNA and total HIV DNA levels in (**A**) COBRA and (**B**) AIMS cohorts. Effect sizes and 95% confidence intervals for US RNA are plotted as $\log_{10}$ copies per microgram of total cellular RNA and for total DNA as $\log_{10}$ copies per million peripheral blood mononuclear cells (PBMC). Effect sizes were obtained by fitting

*Figure 2 continued on next page*

*Figure 2 continued*

generalized linear models. Variables associated with HIV RNA or DNA with p-values <0.1 in the univariable analyses were included in the multivariable analyses.

The online version of this article includes the following figure supplement(s) for figure 2:

**Figure supplement 1.** Regression analyses to identify variables associated with cell-associated HIV US RNA and total HIV DNA levels in the AIMS cohort, taking into account either (**A**) duration of cumulative virological suppression on ART or (**B**) duration of current ART regimen, prior to the measurements.

**Figure supplement 2.** Regression analyses to identify variables associated with cell-associated HIV US RNA and total HIV DNA levels in the AIMS cohort, taking into account both duration of continuous virological suppression on ART and duration of current ART regimen, prior to the measurements.

remained significantly lower in NNRTI- than in PI-treated participants (p = 0.006), while a trend in the same direction was observed for total HIV DNA (p = 0.05) (*Figure 3B*). To confirm that the effects of the ART regimen were independent of the duration of virological suppression, we assigned the participants into four groups according to the duration of continuous suppression (0–1 years, 2–5 years, 6–9 years, and 10 years or more) and compared CA RNA and DNA between NNRTI- and PI-treated individuals in every group separately (*Figure 3—figure supplement 2*). In every group, CA RNA levels were lower in NNRTI- than in PI-treated participants, with a similar but weaker effect observed for CA DNA, in complete agreement to the results obtained in the total cohort.

Next, we assessed the associations of individual drugs with CA HIV RNA and DNA levels (*Figure 3—figure supplement 3*). As the vast majority of NNRTI-treated participants received either efavirenz or nevirapine, we wondered whether these two drugs had a similar effect on CA RNA and DNA. To this end, we compared the HIV markers between these two drugs and PIs (*Figure 3C*). While no difference was observed in CA RNA or total DNA levels between efavirenz- and nevirapine-treated participants, CA RNA was significantly lower in participants treated with either of these drugs compared to PI-treated participants, and a trend in the same direction was observed for total DNA. Finally, no differences were observed in either CA RNA or total DNA levels between three individual ritonavir-boosted PIs that were used by the majority of PI-treated participants (atazanavir, darunavir, and lopinavir) (*Figure 3D*). These results demonstrate that the effects of ART regimens on CA RNA and DNA levels were ART class-specific and not drug-specific.

## Discussion

In this study, we demonstrated in two independent cohorts of individuals on suppressive ART that NNRTI-based triple ART regimens are associated with lower levels of CA HIV RNA compared with PI-based regimens. To the best of our knowledge, this is the largest study comparing CA HIV RNA levels between individuals on different ART regimens. Although several studies have compared residual viremia between ART regimens and most have found lower levels in NNRTI-treated than in PI-treated individuals (*Darcis et al., 2020b*), very few groups included other HIV reservoir markers in such comparisons. *Nicastri et al., 2008* reported lower total HIV DNA levels in individuals treated with PI-based regimens and *Sarmati et al., 2012* found no difference in the HIV DNA level by regimen, despite the fact that both studies reported higher residual viremia in PI-treated individuals. *Kiselinova et al., 2015* performed a matched case-control study comparing nevirapine and PIs for residual viremia, total and episomal HIV DNA, and CA HIV RNA and did not find any differences by regimen for any of these markers. Notably, the latter study matched participants for the duration of PI-based or nevirapine-based regimens, but despite this, significant differences were still observed between the nevirapine- and PI-treated groups in total ART duration and duration of plasma HIV RNA suppression. In our study, we reasoned that, in the absence of a priori knowledge of the factors associated with CA HIV levels, such matching, albeit potentially reducing confounding, could introduce a selective bias and the results would therefore not be representative of the total ART-treated population. Instead, we chose for a cohort study design, controlling for a number of factors in the multivariable models.

Among all demographic and clinical variables, only duration of current regimen and duration of virological suppression were associated with the ART regimen, being longer in NNRTI- than in PI-

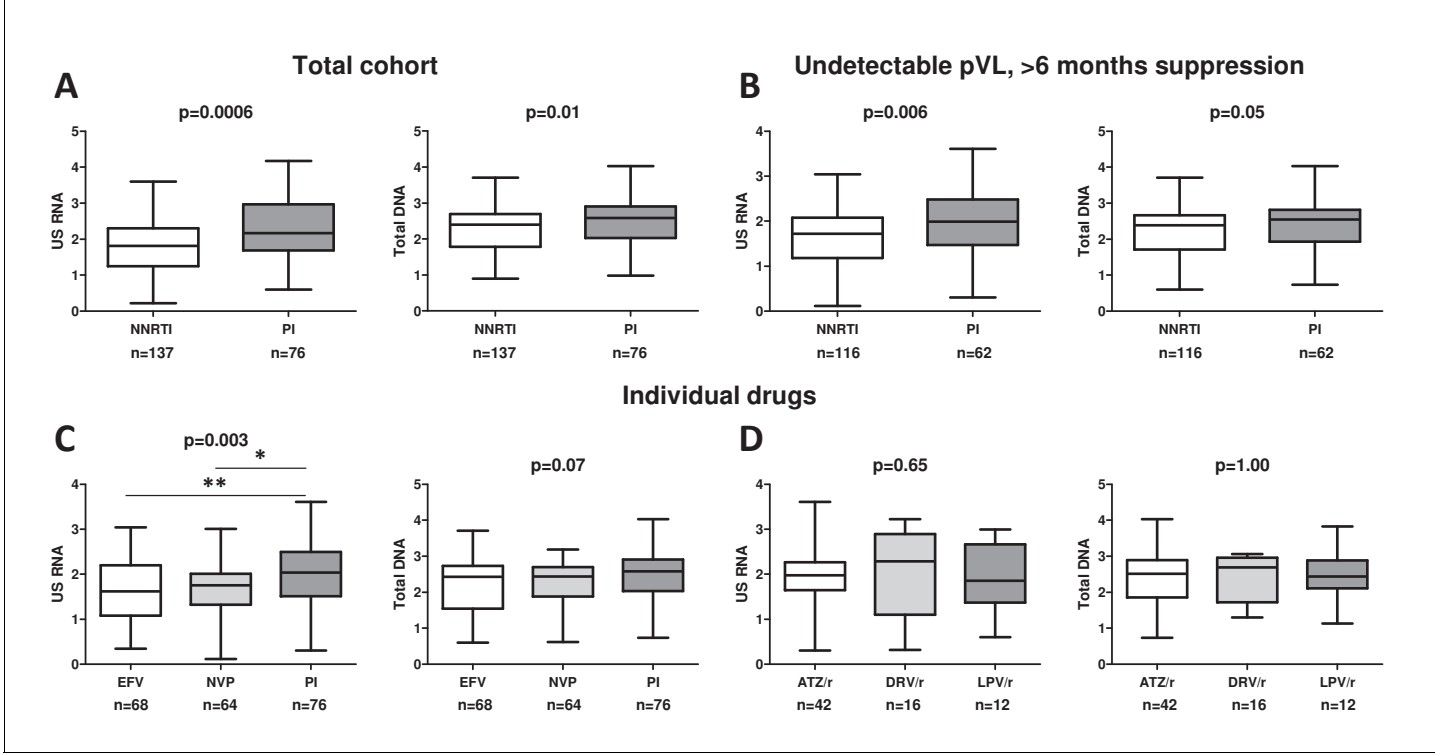

**Figure 3.** Sensitivity analysis and associations of individual antiretroviral drugs with cell-associated human immunodeficiency virus (HIV) RNA and DNA in the pooled cohort. Associations of antiretroviral therapy (ART) regimens (non-nucleoside reverse transcriptase inhibitor (NNRTI)-based vs protease inhibitor (PI)-based) with the levels of cell-associated HIV unspliced RNA (US RNA) and total HIV DNA in either (A) the total pooled cohort (n = 213) or (B) limiting the analysis to participants with undetectable plasma viral loads (pVLs) and >6 months of virological suppression on ART (n = 178). (C) Differences in the levels of US RNA and total HIV DNA between participants treated with ART regimens based on efavirenz (EFV), nevirapine (NVP), or PIs in the total pooled cohort. (D) Differences in the levels of US RNA and total HIV DNA between participants treated with ART regimens based on different ritonavir-boosted PIs: atazanavir (ATZ/r), darunavir (DRV/r), or lopinavir (LPV/r) in the total pooled cohort. Units of measurement are US RNA: $\log_{10}$ copies/µg total RNA, total DNA: $\log_{10}$ copies/$10^6$ peripheral blood mononuclear cells (PBMC). Levels of significance were calculated by Mann-Whitney tests or Kruskal-Wallis tests with Dunn's post-tests, as appropriate. For three-group comparisons, Kruskal-Wallis p-values are shown on top of the graphs and Dunn's significance levels of pairwise comparisons are shown by asterisks only where significant; **$0.001 < p < 0.01$; *$0.01 < p < 0.05$. Participant numbers per regimen are indicated below the graphs.

The online version of this article includes the following figure supplement(s) for figure 3:

**Figure supplement 1.** Association of ART regimen (NNRTI-based vs PI-based) with the cell-associated HIV US RNA/total HIV DNA ratio in the total pooled cohort (n = 213).

**Figure supplement 2.** Associations of ART regimen (NNRTI-based vs PI-based) with the levels of cell-associated HIV US RNA and total HIV DNA in the total pooled cohort.

**Figure supplement 3.** Levels of US RNA and total HIV DNA in participants treated with ART regimens based on efavirenz (EFV), nevirapine (NVP), ritonavir-boosted atazanavir (ATZ/r), ritonavir-boosted darunavir (DRV/r), or ritonavir-boosted lopinavir (LPV/r) in the total pooled cohort.

treated individuals. This can be explained by the fact that while efavirenz and nevirapine, the drugs used by the absolute majority of our NNRTI-treated individuals, were approved for medical use in the late 1990s, modern PIs like atazanavir and darunavir, which were used by most of our PI-treated individuals, were only approved in the mid- or late 2000s. This means that most of the NNRTI-treated individuals in this study started ART, or switched to NNRTIs from the first-generation PIs, earlier than the PI-treated individuals. Because the HIV reservoir generally decays with time on ART (*Siliciano et al., 2003*), this association of duration of suppression with the ART regimen could have potentially confounded the association of the ART regimen with the HIV reservoir measures such as CA RNA and DNA. Indeed, we found that both these HIV markers were negatively associated with the duration of suppression in the AIMS cohort. Interestingly, low-level plasma HIV RNA detectability in that cohort was also negatively associated with the duration of suppression (*Figure 1—figure supplement 2*), confirming the results of a previous study (*Darcis et al., 2020a*). However, to our surprise, no association of CA RNA or DNA with the duration of suppression was found in the COBRA

cohort. One possible reason for this difference between the two cohorts is that the COBRA participants were on average much longer on ART than the AIMS participants (117.8 vs 47.1 months of cumulative virological suppression, respectively). Decay of the HIV reservoir after ART initiation is multiphasic (*Perelson et al., 1997*; *Blankson et al., 2000*), and while the long-term dynamics of CA RNA has not yet been studied in detail, reports on the dynamics of total HIV DNA and residual plasma viremia have demonstrated that these markers reach a plateau after 5–7 years of treatment (*Palmer et al., 2008*; *Besson et al., 2014*; *Bachmann et al., 2019*), possibly due to clonal expansion of the viral reservoir cells (*Maldarelli et al., 2014*; *Wagner et al., 2014*). If the same applies to CA RNA, then it may be expected that after several years on ART, this reservoir measure will also no longer depend on the time on therapy, something that we indeed observed in the COBRA cohort. Instead, in the COBRA cohort, CA RNA and especially total DNA positively correlated with the plasma HIV RNA zenith, suggesting that even after a decade of ART, the HIV reservoir size is still partly determined by its pre-therapy values. Interestingly, most proviral DNA sequences from ART-treated individuals were recently shown to match circulating HIV variants detected shortly before the start of therapy, suggesting that the HIV reservoir quickly turns over in the untreated infection and that the reservoir that persists on ART has been primarily established at the start of therapy (*Abrahams et al., 2019*; *Brodin et al., 2016*; *Pankau et al., 2020*).

Another variable that could have confounded the association of the ART regimen with the HIV reservoir measures such as CA HIV RNA is the adherence to ART. Earlier studies have reported lower adherence among individuals treated with older PI-based ART regimens (*O'Connor et al., 2013*). However, we and others have not previously observed a difference in adherence between individuals treated with modern PI- and NNRTI-based regimens (*Pasternak et al., 2015*; *Konstantopoulos et al., 2015*), and in this study, adherence was also not associated with the therapy regimen. We have previously shown that modest non-adherence correlates with longitudinal changes in CA RNA (*Pasternak et al., 2012*). However, no significant association between adherence and CA RNA levels was observed in this study. Adherence was also not associated with low-level plasma HIV RNA detectability (*Figure 1—figure supplement 2*). The relation between adherence and CA HIV RNA or residual viremia is undoubtedly complex and deserves further research, but it must be noted that while in our previous report the adherence was measured over 1-week periods immediately prior to the HIV sampling moments, in this study we used 1-month adherence measurements taken within 20 days from the sampling moments. Whether short-term adherence is more strongly associated with CA RNA levels or residual viremia remains to be studied.

Notwithstanding the associations of CA RNA with other factors, in both cohorts, NNRTI-based ART was independently associated with lower CA RNA levels as compared with PI-based ART, as revealed by the multivariable analysis. This analysis revealed very similar effect sizes of the ART regimen on CA RNA in both cohorts, despite the fact that several factors, such as duration of virological suppression and the PI drugs, differed between the cohorts. On average, CA RNA levels were 1.75- to 2-fold lower in the NNRTI-treated participants. This confirms numerous reports that measured lower residual plasma viremia in NNRTI- compared to PI-treated individuals (*Darcis et al., 2020b*). In fact, also in this study, low-level plasma HIV RNA was more frequently detectable in PI-treated than in NNRTI-treated participants, despite no difference in therapy adherence by regimen. Notably, a recent large study of more than 12,000 participants starting ART revealed that PI-treated participants were on average 2.7 times more likely to experience virological failure compared with NNRTI-treated participants (*El Bouzidi et al., 2020*). Moreover, three independent clinical trials of triple ART intensification with raltegravir have previously demonstrated much stronger increases in episomal HIV DNA in PI- compared to NNRTI-treated participants, suggesting that at 'baseline', PI-treated individuals had higher levels of residual replication (*Buzón et al., 2010*; *Hatano et al., 2013a*; *Hatano et al., 2011*). Combined, this prior evidence and the results of this study strongly suggest that NNRTIs are more potent in suppressing HIV residual replication than PIs. Constant low-level viral replication, even if it does not cause the development of drug resistance and therapy failure, could exert continuous pressure on the immune system and cause additional morbidity as a result of persistent immune activation, inflammation, and immunosenescence (*Klatt et al., 2013*; *Massanella et al., 2016*). Several studies have reported excess morbidity and mortality rates in infected ART-treated individuals, compared with the general population (*Schouten et al., 2014*; *Guaraldi et al., 2011*; *Lohse et al., 2007*). Although it is still unclear whether this is due to the

adverse effects of the antiretroviral drugs or due to the residual HIV activity, our results argue that an effort should be made to ensure HIV replication is maximally suppressed during therapy.

Interestingly, some NNRTIs, such as rilpivirine, efavirenz, and etravirine, but not nevirapine, can promote selective apoptosis of infected cells by inducing HIV protease-mediated cytotoxicity (*Trinité et al., 2019*). If these NNRTIs are present in cells that are actively producing viral proteins, they may bind to the reverse transcriptase portion of a newly translated Gag-Pol polyprotein and promote its homodimerization, resulting in premature protease activation. This leads to induction of apoptosis and pyroptosis, as well as CARD8 inflammasome activation (*Figueiredo et al., 2006*; *Jochmans et al., 2010*; *Wang et al., 2021*). However, it is very unlikely that such NNRTI effects could contribute to the levels of CA RNA in our cohorts, for two reasons. First, for this mechanism to work, a cell has to express HIV Gag-Pol, and cells that express HIV proteins without ex vivo stimulation are exceedingly rare in ART-treated individuals (*Pardons et al., 2019*). Second, nevirapine, in comparison with other NNRTIs, has no such activity (*Trinité et al., 2019*; *Figueiredo et al., 2006*; *Wang et al., 2021*), but in our study, nevirapine was associated with the same levels of CA RNA as efavirenz, and another study showed even lower levels of residual plasma viremia in nevirapine-treated compared to efavirenz-treated individuals (*Haïm-Boukobza et al., 2011*). This argues against induction of apoptosis of infected cells being a plausible mechanism behind the more pronounced virological suppression by NNRTIs.

Replenishment of the HIV reservoir by residual virus replication has been proposed as one of the mechanisms of HIV persistence (*Chun et al., 2005*). The association of the ART regimen with total HIV DNA in this study was similar to its association with CA RNA: in both cohorts, levels of total DNA were 1.8-fold lower in the NNRTI-treated participants. This suggests that persistent residual replication in the PI-treated participants may have resulted in a larger viral reservoir. However, this has to be interpreted with caution, as no single marker can at present provide a reliable estimate of the HIV reservoir size (*Sharaf and Li, 2017*). Moreover, different definitions of the HIV reservoir exist. Although total HIV DNA is mostly composed of genetically defective proviruses and thus its measurements overestimate the replication-competent reservoir (*Bruner et al., 2016*), these defective proviruses can be transcribed, translated, and even produce defective viral particles, and therefore can contribute to chronic immune activation and inflammation despite ART (*Pollack et al., 2017*; *Imamichi et al., 2020*; *Imamichi et al., 2016*; *Finzi et al., 2006*). Therefore, it has been proposed to extend the definition of the reservoir to all infected cells that can contribute to the residual HIV pathogenesis (*Avettand-Fènoël et al., 2016*; *Baxter et al., 2018*). Furthermore, both total HIV DNA and CA HIV RNA have been shown to predict the time to viral rebound after ART interruption (*Williams et al., 2014*; *Li et al., 2016*), and we have recently reported that CA HIV RNA was predictive of both time to and magnitude of viral rebound after interruption of temporary ART initiated during primary HIV infection (*Pasternak et al., 2020*). This argues that despite being partially composed of defective proviruses, the transcription-competent reservoir does reflect the replication-competent reservoir (*Baxter et al., 2018*; *Abdel-Mohsen et al., 2020*). In view of our present results, future studies should investigate the effects of different ART regimens on the replication-competent reservoir, as the latter is the main obstacle for the development of an HIV cure (*Pasternak and Berkhout, 2016*). Interestingly, *Li et al., 2016* demonstrated that NNRTI-treated individuals experienced a significantly longer time to viral rebound after ART interruption. This may suggest lower replication-competent reservoirs in individuals treated with NNRTI-based ART regimens, although longer half-lives of NNRTIs, leading to prolonged NNRTI exposure after treatment interruption, provide a plausible additional explanation.

Our study has some limitations. First, we did not include individuals treated with INSTI-based ART, which is currently recommended as the first-line therapy (https://aidsinfo.nih.gov/guidelines/html/1/adult-and-adolescent-arv/11/what-to-start), because such individuals were rare (<5%) in the COBRA cohort and absent from the AIMS cohort. Studies in more recent cohorts are necessary to elucidate the differences in CA HIV markers between INSTI-based ART and other regimens. It must be noted that although NNRTI- and PI-based ART regimens are no longer recommended as the first-line therapy in all settings, millions of infected individuals are still treated with these regimens. Thus, our results are relevant for the clinical management of these individuals. Second, although our results provide strong evidence that individuals treated with PI-based ART undergo residual HIV replication, this evidence remains indirect as the direct demonstration of infection of new cells in an ART-treated individual is extremely difficult if not impossible. Therefore, although

studies have used various approaches to prove or disprove the existence of residual replication on ART (reviewed in *Pasternak et al., 2013*; *Martinez-Picado and Deeks, 2016*), all these approaches have so far been indirect. Third, although we adjusted our models for a number of clinical parameters, the observational nature of these cohorts means that residual confounding cannot be entirely excluded. For instance, PI-based regimens could have been preferentially prescribed to individuals with a worse viro-immunological profile and/or expected poor therapy adherence, because PIs impose a relatively high genetic barrier to resistance and consequently will be more 'forgiving' to non-adherence. However, we did not observe any significant differences by regimen in plasma HIV RNA zenith, current and nadir CD4+ count, or therapy adherence, arguing against such a 'prescription bias'. In this regard, a clinical trial with a factorial design, in which participants would switch from PI-based to NNRTI-based ART regimens and vice versa, would be important to confirm our findings.

In summary, here we have demonstrated in two independent cohorts that levels of HIV reservoir markers are lower in individuals treated with NNRTI- as compared to PI-based combination ART. We have previously proposed CA RNA as a sensitive marker of the active HIV reservoir and residual replication (*Pasternak et al., 2013*), a role that is further reinforced by the results of this study. Monitoring of CA RNA levels to detect residual HIV activity despite ART is warranted in order to prevent possible ART complications such as persistent immune activation or therapy failure (*Pasternak et al., 2009*; *El Bouzidi et al., 2020*; *Hatano et al., 2013b*), especially in individuals treated with PI-based regimens.

## Acknowledgements

We are thankful to Gilles Darcis for helpful discussions. We would like to thank the COBRA and AIMS study groups and study participants for helping to establish these cohorts. The COBRA project has received funding from the European Union's Seventh Framework Programme for research, technological development, and demonstration under grant agreement no. 305522. AOP is supported by the grant no. 09120011910035 from the Dutch Medical Research Council (ZonMw).

## Additional information

### Group author details

**The Co-morBidity in Relation to Aids (COBRA) Collaboration**
P Reiss: Department of Global Health and Amsterdam Institute for Global Health and Development (AIGHD), Amsterdam UMC, Universiteit van Amsterdam, Amsterdam, The Netherlands; FWNM Wit: Department of Global Health and Amsterdam Institute for Global Health and Development (AIGHD), Amsterdam UMC, Universiteit van Amsterdam, Amsterdam, The Netherlands; J Schouten: Department of Global Health and Amsterdam Institute for Global Health and Development (AIGHD), Amsterdam UMC, Universiteit van Amsterdam, Amsterdam, The Netherlands; KW Kooij: Department of Global Health and Amsterdam Institute for Global Health and Development (AIGHD), Amsterdam UMC ,Universiteit van Amsterdam, Amsterdam, The Netherlands; RA van Zoest: Department of Global Health and Amsterdam Institute for Global Health and Development (AIGHD), Amsterdam UMC, Universiteit van Amsterdam, Amsterdam, The Netherlands; BC Elsenga: Department of Global Health and Amsterdam Institute for Global Health and Development (AIGHD), Amsterdam UMC, Universiteit van Amsterdam, Amsterdam, The Netherlands; FR Janssen: Department of Global Health and Amsterdam Institute for Global Health and Development (AIGHD), Amsterdam UMC, Universiteit van Amsterdam, Amsterdam, The Netherlands; M Heidenrijk: Department of Global Health and Amsterdam Institute for Global Health and Development (AIGHD), Amsterdam UMC, Universiteit van Amsterdam, Amsterdam, The Netherlands; W Zikkenheiner: Department of Global Health and Amsterdam Institute for Global Health and Development (AIGHD), Amsterdam UMC, Universiteit van Amsterdam, Amsterdam, The Netherlands; T Booiman: Department of Experimental Immunology, Amsterdam UMC, Universiteit van Amsterdam, Amsterdam, The Netherlands; AM Harskamp-Holwerda: Department of Experimental Immunology, Amsterdam

UMC, Universiteit van Amsterdam, Amsterdam, The Netherlands; I Maurer: Department of Experimental Immunology, Amsterdam UMC, Universiteit van Amsterdam, Amsterdam, The Netherlands; MM Mangas Ruiz: Department of Experimental Immunology, Amsterdam UMC, Universiteit van Amsterdam, Amsterdam, The Netherlands; AF Girigorie: Department of Experimental Immunology, Amsterdam UMC, Universiteit van Amsterdam, Amsterdam, The Netherlands; E Frankin: Department of Medical Microbiology, Amsterdam UMC, Universiteit van Amsterdam, Amsterdam, The Netherlands; AO Pasternak: Department of Medical Microbiology, Amsterdam UMC, Universiteit van Amsterdam, Amsterdam, The Netherlands; B Berkhout: Department of Medical Microbiology, Amsterdam UMC, Universiteit van Amsterdam, Amsterdam, The Netherlands; T van der Kuyl: Department of Medical Microbiology, Amsterdam UMC, Universiteit van Amsterdam, Amsterdam, The Netherlands; P Portegies: Department of Neurology, Amsterdam UMC, Universiteit van Amsterdam, Amsterdam, The Netherlands; BA Schmand: Department of Neurology, Amsterdam UMC, Universiteit van Amsterdam, Amsterdam, The Netherlands; GJ Geurtsen: Department of Neurology, Amsterdam UMC, Universiteit van Amsterdam, Amsterdam, The Netherlands; JA ter Stege: Department of Neurology, Amsterdam UMC, Universiteit van Amsterdam, Amsterdam, The Netherlands; M Klein Twennaar: Department of Neurology, Amsterdam UMC, Universiteit van Amsterdam, Amsterdam, The Netherlands; CBLM Majoie: Department of Radiology, Amsterdam UMC, Universiteit van Amsterdam, Amsterdam, The Netherlands; MWA Caan: Department of Radiology, Amsterdam UMC, Universiteit van Amsterdam, Amsterdam, The Netherlands; T Su: Department of Radiology, Amsterdam UMC, Universiteit van Amsterdam, Amsterdam, The Netherlands; K Weijer: Department of Cell Biology, Amsterdam UMC, Universiteit van Amsterdam, Amsterdam, The Netherlands; A Kalsbeek: Department of Experimental neuroendocrinology, Amsterdam UMC, Universiteit van Amsterdam, Amsterdam, The Netherlands; M Wezel: Department of Ophthalmology, Amsterdam UMC, Universiteit van Amsterdam, Amsterdam, The Netherlands; HG Ruhé: Department of Psychiatry, Amsterdam UMC, Universiteit van Amsterdam, Amsterdam, The Netherlands; C Franceschi: Department of Experimental, Diagnostic and Specialty Medicine, Alma Mater Studiorum Universita di Bologna, Bologna, Italy; P Garagnani: Department of Experimental, Diagnostic and Specialty Medicine, Alma Mater Studiorum Universita di Bologna, Bologna, Italy; C Pirazzini: Department of Experimental, Diagnostic and Specialty Medicine, Alma Mater Studiorum Universita di Bologna, Bologna, Italy; M Capri: Department of Experimental, Diagnostic and Specialty Medicine, Alma Mater Studiorum Universita di Bologna, Bologna, Italy; F Dall'Olio: Department of Experimental, Diagnostic and Specialty Medicine, Alma Mater Studiorum Universita di Bologna, Bologna, Italy; M Chiricolo: Department of Experimental, Diagnostic and Specialty Medicine, Alma Mater Studiorum Universita di Bologna, Bologna, Italy; S Salvioli: Department of Experimental, Diagnostic and Specialty Medicine, Alma Mater Studiorum Universita di Bologna, Bologna, Italy; J Hoeijmakers: Department of Genetics, Erasmus Universitair Medisch Centrum Rotterdam, Rotterdam, The Netherlands; J Pothof: Department of Genetics, Erasmus Universitair Medisch Centrum Rotterdam, Rotterdam, The Netherlands; M Prins: Cluster of Infectious Diseases, research department, GGD Amsterdam/Public Health Service Amsterdam, Amsterdam, The Netherlands; M Martens: Cluster of Infectious Diseases, research department, GGD Amsterdam/Public Health Service Amsterdam, Amsterdam, The Netherlands; S Moll: Cluster of Infectious Diseases, research department, GGD Amsterdam/Public Health Service Amsterdam, Amsterdam, The Netherlands; J Berkel: Cluster of Infectious Diseases, research department, GGD Amsterdam/Public Health Service Amsterdam, Amsterdam, The Netherlands; M Totté: Cluster of Infectious Diseases, research department, GGD Amsterdam/Public Health Service Amsterdam, Amsterdam, The Netherlands; S Kovalev: Cluster of Infectious Diseases, research department, GGD Amsterdam/Public Health Service Amsterdam, Amsterdam, The Netherlands; M Gisslén: Göteborgs Universitet, Gotenburg, Sweden; D Fuchs: Göteborgs Universitet, Gotenburg, Sweden; H Zetterberg: Göteborgs Universitet, Gotenburg, Sweden; A Winston: Department of Medicine, Division of Infectious Diseases, Imperial College of Science, Technology and Medicine, London, United Kingdom; J Underwood: Department of Medicine, Division of Infectious Diseases, Imperial College of Science, Technology and Medicine, London, United Kingdom; L McDonald: Department of Medicine, Division of Infectious Diseases, Imperial College of Science, Technology and Medicine, London, United Kingdom; M Stott: Department of Medicine, Division

of Infectious Diseases, Imperial College of Science, Technology and Medicine, London, United Kingdom; K Legg: Department of Medicine, Division of Infectious Diseases, Imperial College of Science, Technology and Medicine, London, United Kingdom; A Lovell: Department of Medicine, Division of Infectious Diseases, Imperial College of Science, Technology and Medicine, London, United Kingdom; O Erlwein: Department of Medicine, Division of Infectious Diseases, Imperial College of Science, Technology and Medicine, London, United Kingdom; N Doyle: Department of Medicine, Division of Infectious Diseases, Imperial College of Science, Technology and Medicine, London, United Kingdom; C Kingsley: Department of Medicine, Division of Infectious Diseases, Imperial College of Science, Technology and Medicine, London, United Kingdom; DJ Sharp: Department of Medicine, Division of Brain Sciences, The Computational, Cognitive & Clinical Neuroimaging Laboratory, Imperial College of Science, Technology and Medicine, London, United Kingdom; R Leech: Department of Medicine, Division of Brain Sciences, The Computational, Cognitive & Clinical Neuroimaging Laboratory, Imperial College of Science, Technology and Medicine, London, United Kingdom; JH Cole: Department of Medicine, Division of Brain Sciences, The Computational, Cognitive & Clinical Neuroimaging Laboratory, Imperial College of Science, Technology and Medicine, London, United Kingdom; S Zaheri: Stichting HIV Monitoring, Amsterdam, The Netherlands; MMJ Hillebregt: Stichting HIV Monitoring, Amsterdam, The Netherlands; YMC Ruijs: Stichting HIV Monitoring, Amsterdam, The Netherlands; DP Benschop: Stichting HIV Monitoring, Amsterdam, The Netherlands; D Burger: Stichting Katholieke Universiteit Nijmegen, Nijmegen, The Netherlands; M de Graaff-Teulen: Stichting Katholieke Universiteit Nijmegen, Nijmegen, The Netherlands; G Guaraldi: Department of Medical and Surgical Sciences for Children & Adults, Università degli studi di Modena e Reggio Emilia, Modena, Italy; T Sindlinger: Department of Biology, Universität Konstanz, Konstanz, Germany; M Moreno-Villanueva: Department of Biology, Universität Konstanz, Konstanz, Germany; A Keller: Department of Biology, Universität Konstanz, Konstanz, Germany; C Sabin: Research Department of Infection and Population Health, University College London, London, United Kingdom; D de Francesco: Research Department of Infection and Population Health, University College London, London, United Kingdom; C Libert: Inflammation research center, Vlaams Instituut voor Biotechnologie, Ghent, Belgium; S Dewaele: Inflammation research center, Vlaams Instituut voor Biotechnologie, Ghent, Belgium; M van der Valk: Division of Infectious Diseases, Amsterdam UMC, Universiteit van Amsterdam, Amsterdam, The Netherlands; NA Kootstra: Department of Experimental Immunology, Amsterdam UMC, Universiteit van Amsterdam, Amsterdam, The Netherlands; J Villaudy: Department of Medical Microbiology, Amsterdam UMC, Universiteit van Amsterdam, Amsterdam, The Netherlands; PHLT Bisschop: Division of Endocrinology and Metabolism, Amsterdam UMC, Universiteit van Amsterdam, Amsterdam, The Netherlands; I Visser: Department of Psychiatry, Amsterdam UMC, Universiteit van Amsterdam, Amsterdam, The Netherlands; A Bürkle: Department of Biology, Universität Konstanz, Konstanz, Germany

## Funding

| Funder | Grant reference number | Author |
|--------|------------------------|--------|
| ZonMw | 09120011910035 | Ben Berkhout |
| FP7 Health | 305522 | Peter Reiss |

The funders had no role in study design, data collection and interpretation, or the decision to submit the work for publication.

## Author contributions

Alexander O Pasternak, Conceptualization, Formal analysis, Investigation, Methodology, Writing - original draft, Writing - review and editing; Jelmer Vroom, Investigation; Neeltje A Kootstra, Marijn de Bruin, Davide De Francesco, Alan Winston, Data curation; Ferdinand WNM Wit, Conceptualization, Methodology; Margreet Bakker, Resources, Data curation; Caroline A Sabin, Formal analysis, Methodology, Writing - review and editing; Jan M Prins, Data curation, Project administration, Writing - review and editing; Peter Reiss, Data curation, Funding acquisition, Project administration, Writing - review and editing; Ben Berkhout, Conceptualization, Supervision, Funding acquisition, Writing

- review and editing; The Co-morBidity in Relation to Aids (COBRA) Collaboration, Funding acquisition

### Author ORCIDs

Alexander O Pasternak https://orcid.org/0000-0002-4097-4251
Neeltje A Kootstra http://orcid.org/0000-0001-9429-7754

### Ethics

Human subjects: The COBRA study was approved by the institutional review board of the Academic Medical Center (Medisch Ethische Toetsingscommissie, reference number NL 30802.018.09) and a UK Research Ethics Committee (REC) (reference number 13/LO/0584 Stanmore, London). All participants provided written informed consent. The AIMS study was approved by the institutional review board of the Academic Medical Center (protocol number NTR176). The trial is registered at https://www.isrctn.com (ISRCTN97730834). All participants provided written informed consent.

### Decision letter and Author response

Decision letter https://doi.org/10.7554/eLife.68174.sa1
Author response https://doi.org/10.7554/eLife.68174.sa2

## Additional files

### Supplementary files

• Supplementary file 1. Variables associated with cell-associated human immunodeficiency virus (HIV) unspliced (US) RNA and total HIV DNA levels in both cohorts. (**a**) Variables associated with cell-associated HIV US RNA and total HIV DNA levels in the COmorBidity in Relation to AIDS (COBRA) cohort. (b) Variables associated with cell-associated HIV US RNA and total HIV DNA levels in the Adherence Improving Self-Management Strategy (AIMS) cohort.

• Transparent reporting form

### Data availability

All data generated or analysed during this study are included in the manuscript and supporting files.

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
