## [Decision Letter]

**Acceptance summary:**

This study examines measures of the HIV reservoir in two cohorts treated with different ART regimens. The findings indicate that the HIV reservoir is lower in virally suppressed individuals on NNRTI-based than PI-based regimens. This was observed in both cohorts, further supporting this finding.

**Decision letter after peer review:**

Thank you for submitting your article "Non-nucleoside reverse transcriptase inhibitor-based combination antiretroviral therapy is associated with lower cell-associated HIV RNA and DNA levels as compared with therapy based on protease inhibitors" for consideration by *eLife*. Your article has been reviewed by 3 peer reviewers, including Julie M Overbaugh as the Reviewing Editor and Reviewer #1, and the evaluation has been overseen by Jos van der Meer as the Senior Editor.

Essential revisions:

This is a well-written study that will be of interest to many investigators working in the field of HIV persistence during ART. The strengths of the study include the analysis of samples from two relatively large cohorts of individuals (n = 100 and 124) and the use of multivariable models to adjust for numerous parameters. One weakness is the fact that the authors do not consider alternative models that may explain their results. This includes the interaction between viral reservoir and longer viral suppression, both of which were associated with NNRTI-based regimens. The data is important and shows that NNRTI usage is associated with lower levels of HIV persistence markers, but it does not provide a mechanistic explanation for that. Overall, this is a well-conducted and important study, with new findings that have potential clinical implications.

1) A main concern is the interaction between duration of viral suppression and the reservoir given the reservoir decays with time. This could contribute to the results and appears to do so for the AIMs cohort; it feels like it is a bit of a chicken and egg issue with this which probably should be discussed more in terms of limitations. The authors do one analysis where they restrict the cases to those with more the 6 months of viral suppression to try to get at the impact of suppression. But I think this would be more telling if they did an analysis around intervals of suppression (6 months to 1 year, 1 year to 2 years etc) as the analysis they did still could be driven by the long duration of some individuals and the associated decay due to that.

2) At the end of the abstract and in a large part of the discussion, the authors propose that their results are explained by the fact that NNRTI have a greater ability to suppress HIV replication. There are alternative explanations: It may be that people on NNRTI clear their transcriptionally active reservoir continuously: The authors should discuss their findings in connection with the recent study by the group of Dr Shan (Lang et al. Nature 2021). In this study, the authors demonstrated that the use of NNRTI may lead to the clearance of transcriptionally active HIV-infected cells through autophagy. Interestingly, the use of PI would block this phenomenon. Lang et al. found that both efavirenz and rilpivirine can lead to the death of transcriptionally active HIV-infected cells, whereas nevirapine has no effect (because it does not enhance dimerization of the gag-pol polyprotein intracellularly). Although this contrasts with the results of the present study, this should be extensively discussed. Also, although the number of participants receiving rilpivirine is probably small, they may be included Figure 3C.

3) The authors should perform a more exhaustive review of the literature in the introduction by quoting the studies that have compared the effects of different ART regimens on HIV DNA and cell-associated HIV RNA levels. See for example Nicastri et al., Curr HIV Res 2008. Some of these studies are included in the discussion, but there are probably many more that should be included in the introduction (including comparisons with integrase inhibitors). I suggest to briefly review these studies after the sentence "However, to date, few studies have compared levels of CA HIV reservoir markers between different ART regimens".

4) Table 1 shows that in the COBRA cohort, the participants on PI regimens have a shorter duration of continuous viral suppression compared to those taking NNRTI. This is likely an important bias here and although the adjusted analysis shows that the differences between the two groups persist after adjusting for this parameter, this should be discussed (page 6 when these differences are presented, not only in the discussion). Same remark for the AIMS cohort.

5) In all correlations, it would be nice to represent the participants PI VS NNRTI using two different colors. This will help the reader determining whether these two groups from separate clusters in all these associations.

6) The differences between Figure 3 A and 3B should be visible in the figure (add a title/label at the top of the graphs or change the Y axis label).

7) Figure 3 C (and D) should be modified: The authors compared each NNRTI (EFV or NVP to all PI in A and B) but each PI to the other PIs only (no comparison with NNRTI). It would be much simpler to redraw this figure and represent all drugs individually (PIs and NNRTIs) and use multiple comparisons.

8) Have the authors attempted to calculate the HIV RNA/HIV DNA ratio in both groups? This would be a better indicator of the viral transcriptional activity.

9) The authors should discuss the need for a clinical trial in which participants would switch from PI to NNRTI to definitely confirm their findings in a longitudinal way.

10) in one of their study cohorts, the authors do not observe an association between duration of ART and cell-associated HIV DNA/RNA levels; this could be due to clonal expansion of viral reservoir cells that amplify and expand viral reservoir size during long-term ART. The authors may wish to discuss this.

11) the authors cite Li et al. (ref 60) to argue that NNRTI treatment is associated with delayed viral rebound after treatment interruption; however, this is likely unrelated to viral reservoir size, and rather caused by altered PK and the longer half-life of NNRTIS relative to PIs.

---

## [Author Response]

Essential revisions:This is a well-written study that will be of interest to many investigators working in the field of HIV persistence during ART. The strengths of the study include the analysis of samples from two relatively large cohorts of individuals (n = 100 and 124) and the use of multivariable models to adjust for numerous parameters. One weakness is the fact that the authors do not consider alternative models that may explain their results. This includes the interaction between viral reservoir and longer viral suppression, both of which were associated with NNRTI-based regimens. The data is important and shows that NNRTI usage is associated with lower levels of HIV persistence markers, but it does not provide a mechanistic explanation for that. Overall, this is a well-conducted and important study, with new findings that have potential clinical implications.

We thank the Editor and the Reviewers for their positive assessment of our study. We now included a discussion of possible alternative explanations of our results (see response to comment 2) and an additional figure showing that the differences in CA RNA and DNA between NNRTI- and PI-treated individuals are independent of the duration of suppression (see response to comments 1 and 4).

1) A main concern is the interaction between duration of viral suppression and the reservoir given the reservoir decays with time. This could contribute to the results and appears to do so for the AIMs cohort; it feels like it is a bit of a chicken and egg issue with this which probably should be discussed more in terms of limitations. The authors do one analysis where they restrict the cases to those with more the 6 months of viral suppression to try to get at the impact of suppression. But I think this would be more telling if they did an analysis around intervals of suppression (6 months to 1 year, 1 year to 2 years etc) as the analysis they did still could be driven by the long duration of some individuals and the associated decay due to that.

We understand the concerns of the reviewer and feel several points need to be clarified. First, the duration of suppression could have potentially confounded the association of the ART regimen with CA RNA and DNA only in the AIMS cohort but not in the COBRA cohort, because only in the AIMS cohort the duration of suppression was associated with CA RNA and DNA. We extensively discuss this finding, as well as possible reasons behind it, in the Discussion. Second, to exclude this potential confounding, we performed multivariable analysis, adjusting for the duration of suppression and other variables, and found an independent association of the ART regimen with CA RNA and DNA in both cohorts. We even built three alternative models for the AIMS cohort, which did not change the results. Therefore, the duration of suppression does not confound the association of the ART regimen with CA RNA and DNA. To illustrate this further, we now performed the analysis suggested by the reviewer, assigning the participants into four groups according to the duration of suppression (Figure 3—figure supplement 2), and comparing CA RNA and DNA between NNRTI- and PI-treated individuals in every group separately. In every of the four groups, CA RNA levels were lower in NNRTI- then in PI-treated participants, with a similar but weaker effect observed for CA DNA, in a complete agreement to the results obtained in the total cohort. This is an additional demonstration that the duration of suppression is not a confounding factor in our analysis.

2) At the end of the abstract and in a large part of the discussion, the authors propose that their results are explained by the fact that NNRTI have a greater ability to suppress HIV replication. There are alternative explanations: It may be that people on NNRTI clear their transcriptionally active reservoir continuously: The authors should discuss their findings in connection with the recent study by the group of Dr Shan (Lang et al. Nature 2021). In this study, the authors demonstrated that the use of NNRTI may lead to the clearance of transcriptionally active HIV-infected cells through autophagy. Interestingly, the use of PI would block this phenomenon. Lang et al. found that both efavirenz and rilpivirine can lead to the death of transcriptionally active HIV-infected cells, whereas nevirapine has no effect (because it does not enhance dimerization of the gag-pol polyprotein intracellularly). Although this contrasts with the results of the present study, this should be extensively discussed. Also, although the number of participants receiving rilpivirine is probably small, they may be included Figure 3C.

The reviewer is referring to the recent paper of Wang et al. (Science 2021; 371:1224). Their findings concerning this NNRTI action are not new, several groups studied this in the past (Trinité Retrovirology 2019, Figueiredo PLOS Pathog 2006, Jochmans Retrovirology 2010, Zerbato Antimicrob Agents Chemother 2017). In essence, some NNRTIs, such as rilpivirine, efavirenz, and etravirine, but not nevirapine, can promote selective apoptosis of infected cells by inducing HIV protease-mediated cytotoxicity. If these NNRTIs are present in cells that are actively producing viral proteins, they may bind to the reverse transcriptase portion of a newly translated Gag-Pol polyprotein and promote its homodimerization, resulting in premature protease activation. This leads to a decrease in virus production and non-specific cleavage of multiple host proteins, including proteins that induce apoptosis/pyroptosis (and, as Wang et al. now show, CARD8 inflammasome activation).

However, we think this alternative explanation for the effects of NNRTIs on CA RNA and DNA that we see in our cohorts is very unlikely, for two reasons. First, for this mechanism to work, a cell has to express HIV Gag-Pol, and cells that express HIV proteins without ex vivo stimulation are exceedingly rare in ART-treated individuals. In fact, Pardons et al. (Plos Pathogens 2019) could not find any Gag-expressing cell in any of their patients unless they reversed latency ex vivo. Second, nevirapine, in comparison with other NNRTIs, has absolutely no such activity – and in our study, nevirapine was associated with the same levels of CA RNA and DNA as efavirenz (Figure 3C). In fact, some studies showed even lower levels of residual plasma viremia in nevirapine-treated compared to efavirenz-treated patients (Haim-Boukobza AIDS 2011), which might be because nevirapine better penetrates some anatomical sanctuaries. Therefore, this mechanism cannot really explain our findings. We discuss this in the revised manuscript (pages 20-21). We did not include rilpivirine in Figure 3C because there were only 2 participants treated with this drug in our study, which is insufficient to draw any conclusions about the effects of this drug.

3) The authors should perform a more exhaustive review of the literature in the introduction by quoting the studies that have compared the effects of different ART regimens on HIV DNA and cell-associated HIV RNA levels. See for example Nicastri et al., Curr HIV Res 2008. Some of these studies are included in the discussion, but there are probably many more that should be included in the introduction (including comparisons with integrase inhibitors). I suggest to briefly review these studies after the sentence "However, to date, few studies have compared levels of CA HIV reservoir markers between different ART regimens".

We would like to draw the reviewer’s attention to our recent review published in Viruses (Darcis et al., Viruses 2020; 12, 489), where we extensively review the available evidence on differences in HIV markers between infected individuals treated with different ART regimens. We performed a very rigorous literature search for this review (see Table 1 in the review). This review is cited in the present manuscript (ref. 28). The absolute majority of such studies measured only residual plasma viremia and only a very few studies measured cell-associated HIV markers: we discuss these studies in the Discussion (including the paper of Nicastri et al. the reviewer is referring to) but we followed the suggestion of the reviewer and now refer to them in the Introduction as well (refs. 29-31).

4) Table 1 shows that in the COBRA cohort, the participants on PI regimens have a shorter duration of continuous viral suppression compared to those taking NNRTI. This is likely an important bias here and although the adjusted analysis shows that the differences between the two groups persist after adjusting for this parameter, this should be discussed (page 6 when these differences are presented, not only in the discussion). Same remark for the AIMS cohort.

Please see the response to the comment 1. Yes, in both cohorts the duration of suppression was lower in PI-treated than in NNRTI-treated participants, but in the COBRA cohort this effect could not even potentially have confounded the association of the ART regimen with CA RNA and DNA, because the duration of suppression was not associated with CA RNA and DNA in this cohort. Such potential confounding could have occurred in the AIMS cohort and we performed an extensive multivariable analysis to rule it out. Moreover, we now performed an additional analysis, in which participants were grouped according to the duration of suppression (Figure 3—figure supplement 2). Combined, our results show that CA RNA and DNA are associated with the ART regimen independently of the duration of suppression.

5) In all correlations, it would be nice to represent the participants PI VS NNRTI using two different colors. This will help the reader determining whether these two groups from separate clusters in all these associations.

We thank the reviewer for this suggestion, which we followed. Figure 1 now shows NNRTI- and PI-treated participants in different colors.

6) The differences between Figure 3 A and 3B should be visible in the figure (add a title/label at the top of the graphs or change the Y axis label).

Thank you, titles added on top of the graphs.

7) Figure 3 C (and D) should be modified: The authors compared each NNRTI (EFV or NVP to all PI in A and B) but each PI to the other PIs only (no comparison with NNRTI). It would be much simpler to redraw this figure and represent all drugs individually (PIs and NNRTIs) and use multiple comparisons.

Representing all drugs individually results in comparing too many groups with very unequal sizes (e.g. 68 participants were treated with efavirenz but only 12 with lopinavir), which considerably reduces the statistical power. In addition, we wanted to check whether both EFV and NVP are associated with lower CA RNA levels compared to PIs (and whether there is a difference between EFV and NVP), to rule out the alternative explanation for our results (see comment 2). Therefore, in figure 3C we compare EFV, NVP, and PIs, and in figure 3D we compare different PIs between each other. We think this comparison is more informative, and the results suggest that the effects of ART regimens on the CA RNA and DNA levels are ART class-specific and not drug-specific. However, we now also included the multiple comparisons suggested by the reviewer (Figure 3—figure supplement 3).

8) Have the authors attempted to calculate the HIV RNA/HIV DNA ratio in both groups? This would be a better indicator of the viral transcriptional activity.

We thank the reviewer for this suggestion and now included the comparison of CA RNA/DNA ratios between NNRTI- and PI-treated participants (Figure 3—figure supplement 1). Although there was a trend towards lower CA RNA/DNA ratios in NNRTI-treated participants, the difference was not significant (p=0.19), which is in line with the observations that both CA RNA and DNA levels were significantly lower in NNRTI-treated participants (p=0.0006 and p=0.01, respectively; Figure 3A).

9) The authors should discuss the need for a clinical trial in which participants would switch from PI to NNRTI to definitely confirm their findings in a longitudinal way.

This is added to the Discussion (page 23).

10) in one of their study cohorts, the authors do not observe an association between duration of ART and cell-associated HIV DNA/RNA levels; this could be due to clonal expansion of viral reservoir cells that amplify and expand viral reservoir size during long-term ART. The authors may wish to discuss this.

This is a nice suggestion. We did discuss that the lack of association between duration of suppression and CA RNA and DNA in the COBRA cohort is probably due to the fact that after several years on ART, CA RNA and DNA reach a plateau and do not decline further. The reviewer provides a plausible explanation for this plateau effect. We agree and have included this in the Discussion (page 18).

11) the authors cite Li et al. (ref 60) to argue that NNRTI treatment is associated with delayed viral rebound after treatment interruption; however, this is likely unrelated to viral reservoir size, and rather caused by altered PK and the longer half-life of NNRTIS relative to PIs.

We agree that this could contribute to the results of Li et al. and included this in the Discussion (page 22).